# Content Moderator Mental Health and Associations with Coping Styles: Replication and Extension of Previous Studies

**DOI:** 10.3390/bs15040487

**Published:** 2025-04-08

**Authors:** Ruth Spence, Jeffrey DeMarco

**Affiliations:** Centre for Abuse and Trauma Studies, Middlesex University, London NW4 4BT, UK; r.spence@mdx.ac.uk

**Keywords:** replication, trust and safety, content moderation, mental health, wellbeing

## Abstract

There is an increasing evidence base that demonstrates the psychological toll of content moderation on the employees that perform this crucial task. Nevertheless, content moderators (CMs) can be based worldwide and have varying working conditions. Therefore, there is a need for studies to be replicated to ensure that the results are robust. The current study used a large sample of commercial CMs employed by an international company to replicate the results from two previous studies that relied on an anonymous online survey. The results pertaining to mental health, wellbeing, and the effectiveness of wellbeing services for this population were mostly replicated. Over a quarter of CMs demonstrated moderate to severe psychological distress, and a quarter were experiencing low wellbeing. Further, the results suggest the potential utility for interventions that increase problem-focussed problem solving, as well as a need for the efficacy of wellbeing services to be evaluated more broadly.

## 1. Introduction

Content moderators (CMs) are the professionals who analyse and remove online content that has been reported by users or artificial intelligence systems as potentially harmful, illegal, or breaking the platform’s policies and guidelines ([11]; [20]). They are required to work across cultural contexts and understand what can at times be coded nuances to make decisions about the suitability of content ([21]; [39]); these decisions often need to be made within approximately 10–30 s ([7]; [15]), sometimes thousands of times over any given shift ([7]; [32]). Therefore, alongside banal content such as spam, CMs can be exposed to high levels of graphic images, videos, and text that feature hate speech, violence, and sexual abuse material ([49]).

There is a large evidence base for more established professions such as police and social workers that demonstrates that exposure to the suffering of others can result in detrimental mental health outcomes such as secondary traumatic stress, vicarious trauma, depression, and anxiety (e.g., [10]; [12]; [18]; [25]). Alongside mental health aspects, workers may also experience burnout ([19]; [38]), as well as increased sickness rates and compassion fatigue ([27]; [33]). A systematic review by [26] ([26]) found there was a high prevalence of secondary trauma in legal professionals who were exposed to clients who have experienced trauma. They also found having a personal history of trauma and higher levels of exposure were associated with secondary trauma.

Alongside this literature, there is a growing evidence base that outlines the potential harm that CMs can also experience. Studies with volunteer CMs, those who moderate community platforms like Twitch or Reddit without payment ([11]), find it is common to experience harassment and abuse ([8]; [31]). They often report experiencing feelings of under-appreciation and guilt ([57]), as well as burnout from the amount of work and their exposure to harmful content, some of which affects their offline experiences ([41]) and can result in people leaving moderation ([31]; [40]). Commercial CMs, those who are paid to moderate platforms, also experience a range of issues from PTSD symptoms ([34]; [39]) to more common mental health problems such as anxiety and depression ([49]). In a study by [46] ([46]) CMs reported experiencing intrusive thoughts and sleep disturbance, as well as feelings of cynicism, desensitisation, and detachment.

The psychological impact of content moderation can be contextualised using established occupational and stress response models. The Job Demands–Resources (JD-R) Model ([5]) states that occupational stress can arise when demands of the job exceed an individual’s available resources. CMs face high job demands, such as repeated exposure to graphic and distressing material, which can lead to burnout, secondary trauma, and poor mental health outcomes. Employer resources, such as workplace wellbeing services, should mitigate these effects, but their effectiveness remains uncertain.

Additionally, the Transactional Model of Stress and Coping ([29]) provides a framework for understanding individual differences in responses to coping responses to stress. Stress outcomes largely depend on how individuals assess a stressor and which coping strategies they use. Problem-focussed coping, where one actively addresses the stressors, is linked to better outcomes, while avoidant coping, such as denial, can worsen psychological distress.

Nevertheless, there has been growing recognition of what has been described as a replication crisis in the social sciences (e.g., [50]; [56]). The assumption of psychological studies is that most statistically significant findings can be replicated using new data ([45]). This ability to reproduce the results in novel datasets then suggests that the results are both reliable and generalisable rather than due to some methodological or sampling quirk ([48]). Theories can then be developed based on the empirical patterns of results taken from numerous studies, enhancing knowledge construction and avenues for future research, which further accumulates new knowledge ([3]).

Ideally, any insights should be confirmed using new data, and this is no less true for trust and safety research, especially given that moderators are a worldwide workforce and that their working conditions vary considerably, from volunteers to direct employees to those employed by outsourced third-party vendors ([11]; [39]). However, a significant amount of research occurs and remains inside private companies. This can mean instead of working collaboratively towards industry-wide best practices on topics like trust and safety, research tends to be somewhat internally focussed, and it is unknown to what degree the results may be industry-wide rather than company-specific. Nevertheless, there are good reasons why companies may want research to stay private; for example, disclosures made through describing the research process or environment may benefit a company’s competition or create negative publicity. For example, in 2014, Facebook released the results of their emotional contagion study where users’ feeds had been manipulated to be either positive or negative to explore the emotional impact of Facebook ([28]). However, this study sparked a wave of publish backlash and criticism for breaching ethical guidelines ([4]). Furthermore, the results may open up companies to legal challenges. This could be particularly problematic for studies that investigate employee mental health. For example, a Spanish court ruled in a case that a CM’s mental health condition was due to work-related issues ([17]), and Meta reached a settlement in a court case where they were accused of failing to protect CMs from psychological injuries resulting from exposure to graphic content ([55]).

Previous studies by [47] ([47]) have found high levels of psychological distress, with over a third (34.6%) of CMs surveyed scoring in the moderate-to-severe range of the CORE-10 and almost half (47.6%) scoring 13 or over, which is associated with having clinical depression ([6]). They also found that over a quarter (27.4%) scored in the low wellbeing range on the Short Warwick–Edinburgh Mental Wellbeing Scale (SWEMWBS; [51]). There was also a dose–response effect between frequency of exposure and psychological distress and secondary trauma, with daily exposure associated with greater scores than exposure occurring less than monthly. An exploration of coping strategies found that the majority used approach-based coping and that this was associated with a reduction in psychological distress and secondary trauma and greater wellbeing, whilst avoidant coping was associated with the opposite. Nonetheless, in both approaches, avoidant coping was measured through behavioural indicators such as smoking or talking to colleagues rather than a standardised measure of coping. Additionally, they found that using wellbeing services was not significantly associated with mental health or wellbeing and service use was not predicted by concerns about confidentiality or the belief that they did not understand the pressure of the role ([47]). However, there may be other perceived benefits that employees garner from wellbeing services beyond symptom reduction.

Both these studies relied on an anonymous, online sample. This therefore raises concerns over the validity and reliability of the data. For one, the population to which the survey was distributed cannot be described; this means that the respondents may be unrepresentative in some meaningful way ([1]). For example, it is impossible to track non-response rates ([2]), and the studies likely suffer from self-selection bias ([52]). Those who are traumatised by their job may have been more motivated to respond. Similarly, online surveys may suffer from coverage limitations ([37]), whereby those who completed the survey may not currently be or have ever been CMs (the target population). These limitations make it harder to make generalisations from the study findings but can be ameliorated through replication ([59]).

This paper seeks to replicate key findings from the previous studies and extend them using a large sample of professional CMs working for an international company. Specifically, we aim to (a) explore rates of psychological distress, secondary trauma, and low wellbeing, (b) determine whether there is a dose–response effect between frequency of exposure and greater symptomatology, (c) establish whether there is an association between wellbeing service use and a reduction in symptomatology and if there are other perceived benefits employees gain from wellbeing services, and (d) explore the relationship between coping and symptoms using a validated measure of coping styles. Specifically, this study explored the following hypotheses:

**H1.** 
*Higher exposure to distressing content will be positively correlated with increased psychological distress and secondary trauma.*


**H2.** *Increased use of wellbeing services will be associated with lower levels of psychological distress and secondary trauma*.

**H3.** *Problem-focussed coping will be negatively correlated with psychological distress and secondary trauma*, *whereas avoidant coping will be positively correlated*.

## 2. Materials and Methods

### 2.1. Participants

The participants were 160 CMs recruited from an international company that provides content moderation services in the entertainment industry. The company employs approximately 200 CMs, so this represents an 80% response rate. The participants completed an anonymous online survey that collected their demographic information (e.g., sex, age, and location); assessed their mental health symptoms by way of the CORE-10, a measure of psychological distress; collected responses using the Secondary and Vicarious Trauma Scale and the Short Warwick–Edinburgh Mental Wellbeing Scale; and assessed their working conditions. Whilst the participants in the sample work and live in diverse geographical regions and speak multiple languages, all the CMs are proficient in reading and speaking English.

The link to the online survey was shared with trust and safety professionals at the company who disseminated it to their frontline CMs. The survey included a range of questions about themselves, their job, and their work environment. This included what sort of distressing content they were exposed to, if they use the wellbeing services available at work (often/occasionally/never), concerns about confidentiality (yes/somewhat/no), and whether the professional understands the pressure of the role (yes/somewhat/no), as well as if they thought the wellbeing service has had an impact (more productive/fewer sick days/improved mental health/feel heard/valued/other). The participants were also asked if they had moved from a customer support agent role to their current role (yes/no) and if their wellbeing had changed as a result (improved/stayed the same/gotten worse). Mental health and wellbeing were measured through the CORE-10, Short Warwick–Edinburgh Mental Wellbeing Scale, and the Secondary and Vicarious Trauma Scale, while coping strategies were measured using the Coping Orientation to Problems Experienced Inventory Brief. Participation was unpaid, anonymous, and voluntary, and all the participants read through an information sheet about this study and gave consent before they began. This study was approved by the university’s psychology department ethics board (Ref: 21657).

### 2.2. Measures

#### 2.2.1. Short Warwick–Edinburgh Mental Wellbeing Scale (SWEMWBS; [51])

The SWEMWS is a seven-item self-report measure that asks about thoughts and feelings over the previous two weeks. Items are scored from 1 ‘none of the time’ to 5 ‘all of the time’, and then scores are summed and transformed into metric scores using the SWEMWBS conversion table. Scores range from 7 to 35, with higher scores indicating greater wellbeing. High wellbeing is indicated by a score of 27.5 and above, whilst low wellbeing is indicated by a score of 19.5 and below. The participants’ scores ranged from 7 to 35 (mean = 22.66, SD = 5.08). The scale demonstrated good internal reliability (α = 0.91).

#### 2.2.2. Clinical Outcomes in Routine Evaluation (Core-10; [6])

The CORE-10 assesses psychological distress over the previous week. It consists of 10 items, which are scored from 0 ‘not at all’ to 4 ‘most or all of the time’ and then summed. The items cover anxiety, depression, trauma, physical problems, functioning, and risk to self. Higher scores indicate higher levels of general psychological distress, with severe psychological distress indicated by scores of 25 and above and the non-clinical range indicated by scores of 10 and below. The participants scored between 0 and 35 (mean = 10.69, SD = 7.21), and the scale showed good reliability (α = 0.81).

#### 2.2.3. Secondary and Vicarious Trauma Scale (SVTS)

The SVTS is a 17-item self-report measure assessing aspects of secondary and vicarious trauma. Items are scored from 1 ‘strongly disagree’ to 5 ‘strongly agree’ and covers thoughts (most people are not trustworthy), emotions (I often feel sad), and intrusion (I find it difficult to separate my work and personal life). Scores are summed with higher scores indicating higher levels of secondary and vicarious trauma. The participants’ scores ranged from 21 to 72 (mean = 43.99, SD = 11.99), and the scale exhibited good reliability (α = 0.88).

#### 2.2.4. Coping Orientation to Problems Experienced Inventory (Brief-COPE; [13])

The Brief-COPE is a 28-item self-report measure of effective and ineffective coping strategies. Each item is scored from 0 ‘I have not been doing this at all’ to 3 ‘I have been doing this a lot’. The scores are then summed across three subscales: problem-focussed coping, emotion-focussed coping, and avoidant coping. Problem-focussed coping is where someone directly addresses the source of a distressing situation using strategies like acting to make the situation better or planning what steps to take. Emotion-focussed coping focusses on reducing emotional stress, often through strategies such as seeking social support. Avoidant coping is characterised by trying to avoid dealing with the stressor through strategies like denial or substance use. The participants’ scores ranged from 8 to 32 (mean = 21.06, SD = 5.43) for problem-focussed coping, 10 to 34 (mean = 22.05, SD = 5.42) for emotion-focussed coping, and 8 to 29 (mean = 14.08, SD = 3.53) for avoidant coping. The internal reliability was α = 0.87 for the problem-focussed coping subscale and α = 0.73 for the emotion-focussed and avoidant coping subscales.

### 2.3. Analysis

Descriptive statistics established the frequency of demographic characteristics, type of content CMs were exposed to, frequency of exposure and to quantify coping strategies employed by CMs. A series of multivariate general linear models (MGLMs) were conducted. MGLMs allow multiple dependent variables to be analysed simultaneously and are suitable for situations where the dependent variables are correlated. The first MGLM used assessed the relationship between frequency of exposure and psychological distress, secondary trauma, and wellbeing. Frequency of exposure was trisected into ‘daily’, ‘weekly/monthly’, and ‘less often’ and entered as a predictor. To control for their effects, sex, location, and duration in role were entered as fixed factors, whilst age was centred and entered as a covariate. The total scores on the CORE-10, SVTS, and WEMWBS were entered as the dependent variables. Due to small numbers, participants from Europe, from South America, or who did not identify as male or female were removed from the MGLM analysis. A second MGLM was conducted to explore the relationship between wellbeing service use, confidentiality concerns, and wellbeing services understanding the pressure involved, with psychological distress, secondary trauma, and wellbeing as dependent variables. The predictor variables were dichotomised into ‘yes/no’, with ‘often’ and ‘occasionally’ recoded as yes and ‘never’ recoded as no for wellbeing service use, and with ‘yes’ and ‘somewhat’ recoded as yes and ‘no’ kept as no for concerns about confidentiality and professional understanding of the pressure of the role. Lastly, the different total scores on the problem-focused, emotion-focused, and avoidant coping style sub-scales were entered as predictors, with the CORE-10, SVTS, and WEMWBS total scores as the dependent variables.

## 3. Results

The majority of the participants were male (63.75%) and located in Asia (62.50%). The moderators were aged between 19 and 48 (mean = 29.68, SD = 5.10) and had most commonly been in the role between 6 months and 2 years (mean = 20.71 months, SD = 19.52). This is compared to the respondents in [47] ([47]), where the minority of the participants were male (46.9%), the average age was 34.3 years (SD = 54.52), and the average duration in the role was 51.76 months (SD = 54.52). The analysis demonstrated that the current sample contained significantly more males (χ^2^ = 13.98, *p* < 0.01), the participants were significantly younger (F = 30.60, *p* < 0.001), and the average time in the role was shorter (F = 12.02, *p* < 0.001). The majority of previous participants were based in Europe (45.1%); therefore, the sample location was also significantly different (χ^2^ = 180.84, *p* < 0.001) (see Table 1). The majority of the participants (51.3%) scored in the low-to-mild range of the CORE-10, with a further 24.4% scoring in the healthy range and almost a quarter (24.4%) scoring in the moderate-to-severe range. There were 54 participants (33.75%) who scored 13 or greater on the CORE-10, suggesting that a third of the sample had symptoms associated with clinical levels of depression. Over a quarter of the participants (28.8%) scored in the low wellbeing range and 13.5% scored in the high range. Psychological distress demonstrated a significant negative relationship with wellbeing (*r* = −0.59, *p* < 0.001) and a significant positive relationship with secondary trauma (*r* = 0.77, *p* < 0.001). Wellbeing was negatively correlated with secondary trauma (*r* = −0.62, *p* < 0.001).

### 3.1. Exposure to Content and Mental Health

Over three-quarters (76.9%) of CMs reported being exposed to hate speech, and over a third reported exposure to humiliation (35.6%) and child sexual abuse material (CSAM; 34.4%).

Most frequently, CMs were exposed to content they found to be distressing less than monthly, but over a quarter (28.7%) were exposed on a daily basis and over a fifth (21.3%) on a weekly basis. The MGLM showed that there was a dose–response effect between the frequency of exposure and psychological distress (F(2, 131) = 6.01, *p* = 0.003) and secondary trauma (F(2, 131) = 8.72, *p* < 0.001) but not wellbeing (F(2, 131) = 1.19, n.s.) (see Figure 1). Bonferroni post hoc tests demonstrated that daily exposure was associated with significantly greater psychological distress than those exposed either weekly/monthly (mean difference = 4.49, *p* = 0.006, 95%CI: 0.59–8.38) or less often (mean difference = 4.68, *p* = 0.006, 95%CI: 1.09–8.27). Similarly, daily exposure was associated with significantly greater secondary trauma than either weekly/monthly (mean difference = 8.68, *p* = 0.003, 95%CI: 2.49–14.87) or less frequent exposure (mean difference = 8.90, *p* < 0.001, 95%CI: 3.19–14.60).

### 3.2. Wellbeing Services and Mental Health

The MGLM demonstrated that using wellbeing services was not significantly associated with psychological distress (F(1, 152) = 3.37, n.s.) or wellbeing (F(1, 152) = 1.29, n.s.), but those who used wellbeing services had significantly lower secondary trauma symptoms (F(1, 152) = 4.46, *p* = 0.036). Concerns about confidentiality were significantly associated with higher psychological distress (F(1, 152) = 28.94, *p* < 0.001), secondary trauma (F(1, 152) = 24.38, *p* < 0.001), and lower wellbeing (F(1, 152) = 8.76, *p* = 0.004). There were no associations between the services understanding the pressure of their role and wellbeing (F(1, 152) = 1.15, n.s.), psychological distress (F(1, 152) = 1.65, n.s.), or secondary trauma (F(1, 152) = 3.04, n.s.) (see Figure 2). An interaction effect between concerns about confidentiality and using wellbeing services was checked but was found not to be significant for any of the outcomes.

In terms of perceived benefits, the majority of the participants (60.6%) reported that the wellbeing service made them feel valued/heard, 40.0% thought it improved their mental health, a third (33.1%) stated that it made them more productive, and almost a tenth (9.4%) reported that they took fewer sick days. Interestingly, of the 89 (55.6%) participants who had moved from a customer support role to content moderation, half (50.0%) asserted it had improved their wellbeing.

### 3.3. Coping Styles and Mental Health

An MGLM showed problem-focussed coping was associated with lower psychological distress (F(1, 152) = 27.26, *p* < 0.001), secondary trauma (F(1, 152) = 23.81, *p* < 0.001), and greater wellbeing (F(1, 152) = 33.38, *p* < 0.001). Emotion-focussed coping was also associated with lower psychological distress (F(1, 152) = 8.99, *p* = 0.003), secondary trauma (F(1, 152) = 9.14, *p* = 0.003), and greater wellbeing (F(1, 152) = 4.01, *p* = 0.047), whilst avoidant coping was associated with greater psychological distress (F(1, 152) = 28.64, *p* < 0.001), secondary trauma (F(1, 152) = 27.35, *p* < 0.001), and lower wellbeing (F(1, 152) = 9.03, *p* = 0.003). An increase in psychological distress scores was associated with increased emotion-focussed coping and avoidant coping, whereas problem-focussed coping was associated with a decrease in psychological distress scores (see Figure 3).

Similarly, an increase in secondary trauma scores was associated with greater emotion-focussed and avoidant coping and less problem-focussed coping (see Figure 4).

Lastly, wellbeing demonstrated the opposite pattern, where it increased with problem-focussed coping and decreased with greater emotion-focussed and avoidant coping (see Figure 5).

## 4. Discussion

This study sought to replicate and extend findings from previous studies investigating CM mental health and wellbeing using a defined sample of CMs. The response rate in the current study was 80%, providing confidence that these results are representative of the CMs currently employed at this company. The current study sample is significantly different on a number of demographic dimensions from the previous sample, yet the results pertaining to mental health are consistent. In both this and the study by [47] ([47]), a large number of CMs (25–33%) scored in the moderate-to-severe range of the CORE-10 and had scores suggesting clinical depression (33–50%). Further, over a quarter of moderators scored in the low wellbeing range. The dose–response effect between the frequency of exposure to content and levels of psychological distress and secondary trauma was replicated, with individuals exposed to content on a daily basis reporting significantly higher psychological distress and secondary trauma symptoms than those exposed less often. Similarly, the lack of a significant association between exposure and wellbeing was reproduced.

Research using the CORE-10 in non-clinical studies is lacking; however, a study by [9] ([9]) found that 17.2% of students scored in the moderate-to-severe range of the CORE-10. A rate lower than that was found in studies on CMs. Additionally, [16] ([16]) found a mean CORE-10 score in the general population (n = 535 adults) of 4.7, indicating that moderators again skew high. Nevertheless 56.9% of moderators in the current sample scored in the healthy-to-low range, 17.58% (n = 16) of whom were exposed to distressing content on a daily basis. Given that wellbeing service use was not significantly associated with psychological distress, this suggests there are individuals who are naturally more resilient to this type of work. Future research should focus on exploring what factors are associated with resilience in this high-stress role, both in terms of screening for features that may make individuals less vulnerable but also what can be learned from resilient individuals to influence wellbeing policy more widely.

The lack of association between using wellbeing services and mental health and wellbeing found in [47] ([47]) was partially replicated. Using the wellbeing services was not significantly associated with psychological distress or wellbeing, but it was associated with reduced secondary trauma. These results align with the JD-R model, reinforcing the idea that excessive job demands contribute to psychological distress and secondary trauma. More frequent exposure was associated with increased psychological distress and secondary trauma symptoms but not wellbeing. Wellbeing services play a valuable role in supporting CMs, particularly in alleviating secondary trauma symptoms, and many CMs reported feeling heard and valued through their engagement with these services. However, the findings suggest that enhancing the structure and targeting these initiatives could further strengthen their impact, particularly in addressing psychological distress and overall wellbeing.

These results highlight the need to formally evaluate “what works” across therapeutic services and the intervention-specific components that leverage change in terms of different issues, whether it be alleviating secondary trauma symptoms or psychological distress more generally. Whereas [47] ([47]) found not understanding the pressure of the role to be associated with higher psychological distress, secondary trauma, and lower wellbeing, there are no significant relationships in the current study. Lastly, the relationships between concerns about confidentiality and psychological distress, secondary trauma, and lower wellbeing were the same. This underlines the importance of assuring CMs that the psychological support offered to them at work is confidential. As before, concerns with confidentiality did not significantly interact with using services; therefore, CMs may still use services but not engage with them in the same way. Future research should try to tease out why CMs are worried about confidentiality and how this drives greater symptomology and reduced wellbeing.

The absence of a significant relationship between exposure to distressing content and overall wellbeing aligns with the previous work with CMs; however, this contrasts with wider research with other populations (e.g., [47]). One possible explanation lies in sample differences. This study’s participants were younger, had shorter tenure, and were predominantly male—factors that may influence resilience and reporting of distress ([42]). Cultural influences may also shape how distress is perceived and reported, particularly given that over 60% of the sample was based in Asia, where emotional restraint in professional settings is often emphasised ([58]). This could contribute to the under-reporting of exposure-related wellbeing impacts. Coping mechanisms may have played a moderating role. The study found that problem-focussed coping was associated with better wellbeing, suggesting that some CMs develop adaptive strategies that protect against the negative effects of exposure. The finding that some moderators reported improved wellbeing after transitioning from customer support roles suggests that job-related factors beyond exposure alone may influence outcomes. Lastly, the WEMWBS assesses general wellbeing rather than specific occupational stressors, which may explain the divergence from previous findings. Future research could employ a broader set of wellbeing measures and explore longitudinal trends.

Nevertheless, beyond reducing symptoms, there were other perceived benefits of having wellbeing services available. Most notably, the majority felt that having it available made them feel heard/valued. This has numerous benefits beyond mental health, including increased job satisfaction and motivation ([35]; [43]) and decreased intention to leave ([30]). Additionally, almost half thought it improved their mental health. The CORE-10 is a measure of psychological distress, and it is possible that other mental health scales may detect a reduction in symptoms. Nonetheless, mental health is not merely the absence of mental illness, and the two are related but distinct ([54]); therefore, wellbeing services may also help to improve mental health without reducing symptoms. Interestingly, half of respondents who moved from a player support role, where the work typically involves responding to players’ queries about their accounts, gameplay, etc., to a moderation role, reported an improvement in their wellbeing. Given the more stressful nature of moderation, this was unexpected. However, although the company provides a general wellbeing service across all teams, the one for its content moderators has been specifically tailored to address their unique challenges, which might explain why self-reported wellbeing improves. This again highlights the need to tease out what components of wellbeing services are working and what different issues they serve to address in content moderators. These results highlight the importance of continuously refining wellbeing interventions to align with the evolving needs of CMs. The JD-R model emphasises the role of job resources in mitigating workplace stress, and this study underscores the opportunity to optimise existing services by tailoring them to the specific challenges faced in CM roles.

While wellbeing services were positively associated with reduced secondary trauma, their impact on broader mental health outcomes could be improved through targeted interventions, such as resilience-building programs or specialised mental health support. This presents a valuable opportunity to build on existing strengths and ensure that interventions are as effective as possible for those who need them. Interventions should include proactive mental health screening, ensuring early support rather than relying solely on self-referrals. Concerns about confidentiality indicate a need for clearer privacy assurances and anonymous access to services to encourage engagement. Given the association between problem-focussed coping and better wellbeing, structured resilience training that teaches adaptive coping strategies could improve CMs’ ability to manage stress. Since many CMs valued wellbeing services despite limited symptom reduction, peer support programs and mentorship initiatives could further strengthen workplace support systems.

The sample was predominantly from Asia and North America, which raises the possibility of cultural influences on coping and wellbeing. Collectivist cultures emphasise emotional restraint, which may contribute to the under-reporting of distress ([58]), while mental health stigma in some regions could explain why confidentiality concerns are associated with higher distress ([14]). Additionally, workplace mental health policies and management attitudes toward wellbeing differ across regions, potentially affecting engagement with wellbeing services ([24]; [44]).

This study used the Brief-COPE scale to extend the previous results concerning coping strategies that CMs use and their effect on mental health and wellbeing. Emotion-focussed and avoidant coping were associated with increased psychological distress, secondary trauma, and decreased wellbeing, whereas problem-focussed coping was associated with the opposite. This is consistent with the previous literature, which shows problem-focussed coping is associated with greater resilience and fewer mental health symptoms, whilst avoidant and emotion-focussed coping are associated with poorer mental health outcomes (e.g., [14]; [23]; [53]). These results show a potential avenue for training that increases problem-focussed problem solving as a wellbeing initiative in trust and safety workplaces. This is consistent with the Transactional Model of Stress and Coping. Problem-focussed coping was associated with lower psychological distress and higher wellbeing, while avoidant coping correlated with worse outcomes. This supports prior research showing that active engagement with stressors mitigates negative effects, whereas disengagement or avoidance can exacerbate psychological distress ([14]; [53]). The results demonstrate the value of training CMs in adaptive problem-solving techniques to increase resilience and counter the effects of chronic exposure to distressing content.

This study highlights that not all workplace resources are equally effective, demonstrating the need for interventions that directly address unique stressors involved with moderation. It also demonstrates that problem-focussed coping mitigates distress while avoidant coping increases it. Some CMs maintained wellbeing despite high exposure, suggesting a need to explore dispositional resilience factors.

To improve workplace mental health outcomes, companies should implement structured, evidence-based interventions tailored to the unique stressors of moderation work. Cognitive Behavioural Therapy programs for CMs could help develop adaptive coping mechanisms, particularly in reducing rumination and emotional disengagement ([36]). Resilience-building initiatives, such as Acceptance and Commitment Therapy, have shown promise in trauma-exposed professions by enhancing emotional regulation and distress tolerance ([22]).

While AI-assisted content filtering is often proposed as a solution to reduce exposure, its effectiveness remains limited, as CMs must still review flagged material manually. Companies should focus on structured exposure management strategies, such as rotating content types and integrating mandatory breaks, which have been shown to mitigate distress in high-exposure roles ([32]).

### Limitations

Although many of the previous findings have been replicated using a new sample—crucially, one that is significantly different in terms of demographics from the original sample—it has not escaped the authors that we are replicating our own work with data that we cannot publicly share. Other researchers need to work with industry partners to publicly replicate these results. Nevertheless, the replication of the original results adds more credence to their generalisability and reliability. This study employed a cross-sectional design, and while this approach allows for a broad assessment, it does not account for how experiences evolve over time. Given that occupational stress can accumulate and that symptoms of secondary trauma may not appear immediately, a longitudinal design would provide stronger insights into how CMs’ mental health changes with prolonged exposure. Future research should prioritise longitudinal studies that follow CMs over time, assessing whether distress levels increase, stabilise, or reduce with experience and identifying factors that contribute to resilience or burnout in the long term.

## 5. Conclusions

The results of the current study mostly replicate previous findings that used a large anonymous online sample. This indicates that the results are generally robust and suggests that approximately a third of CMs are likely to demonstrate raised rates of psychological distress and that a quarter will experience low wellbeing. This, combined with the consistent finding that wellbeing services demonstrate limited effectiveness, highlights the ongoing importance of discovering ‘what works and for whom’ with this population of workers. One potential avenue is exploring training that increases the use of problem-solving-focussed coping strategies. The lack of longitudinal research also urgently needs to be addressed, especially when many of these workers are young and the long-term effects are currently unknown.

## Figures and Tables

**Figure 1 behavsci-15-00487-f001:**
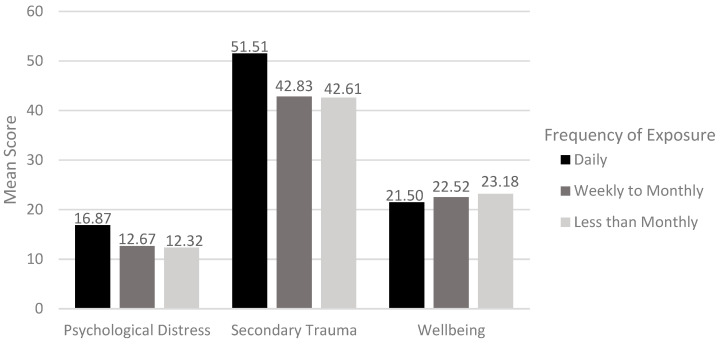
Mean scores on psychological distress, secondary trauma, and wellbeing by frequency of exposure.

**Figure 2 behavsci-15-00487-f002:**
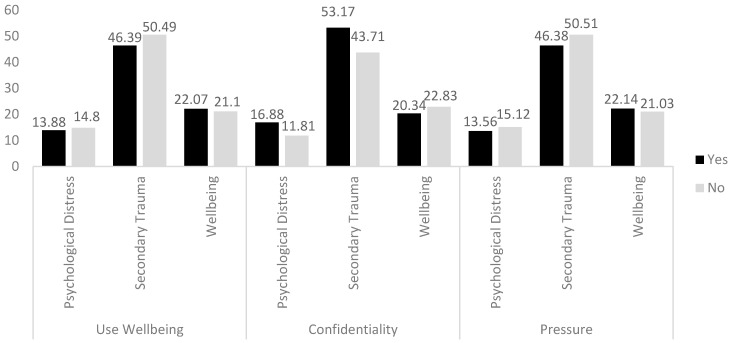
Mean scores on psychological distress, secondary trauma, and wellbeing by wellbeing service use, concerns about confidentiality, and if services understand the pressure of the role.

**Figure 3 behavsci-15-00487-f003:**
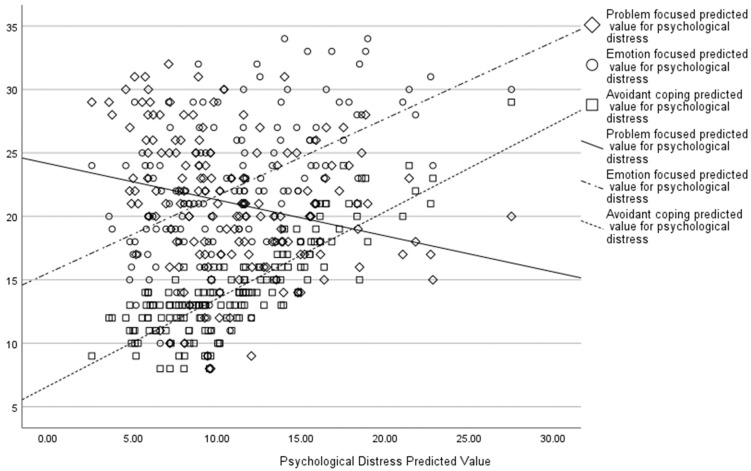
Associations between psychological distress predicted values and different coping strategies.

**Figure 4 behavsci-15-00487-f004:**
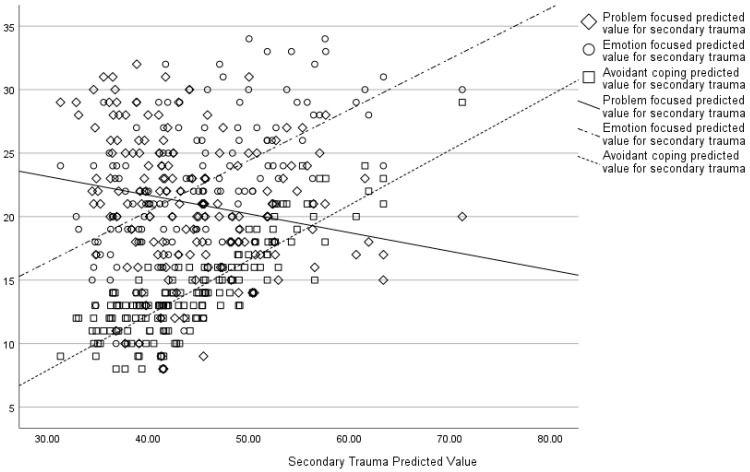
Associations between secondary trauma predicted values and different coping strategies.

**Figure 5 behavsci-15-00487-f005:**
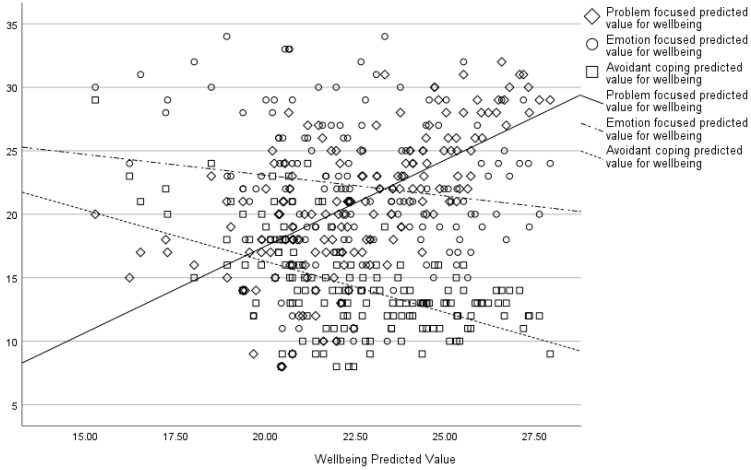
Associations between wellbeing predicted values and different coping strategies.

**Table 1 behavsci-15-00487-t001:** Participant demographics.

Characteristic	Category	Frequency
Sex (n = 160)	Male	102 (63.75)
	Female	47 (29.38)
	Non-binary/other	5 (3.12)
	Declined to answer	6 (3.75)
Location (n = 160)	Asia	100 (62.50)
	North America	57 (35.62)
	Europe	2 (1.25)
	South America	1 (0.62)
Age (n = 157)	19–25	27 (16.8)
	26–35	111 (68.9)
	36+	19 (11.8)
Duration in role (n = 160)	0–6 months	28 (17.5)
	7–12 months	46 (28.7)
	13 months–2 years	47 (29.4)
	2 years, 1 month–5 years	32 (20.0)
	5+ years	7 (4.4)
Exposure (n = 160)	Daily	46 (28.7)
	Weekly	34 (21.3)
	Monthly	9 (5.6)
	Less often	61 (38.1)
	Never	10 (6.3)
Use wellbeing services (n = 159)	Never	50 (31.4)
	Occasionally	82 (51.2)
	Often	27 (16.9)
Concerns about confidentiality (n = 159)	Yes	16 (10.1)
	Somewhat	33 (20.6)
	No	110 (69.2)
Understand the pressure (n = 159)	Yes	59 (37.1)
	Somewhat	71 (44.7)
	No	29 (18.2)

## Data Availability

The datasets presented in this article are not readily available because they came from employees of an international company that does not want the data to be shared. Requests to access the datasets should be directed to Jeffrey DeMarco.

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
