# Peer review of "Content Moderator Mental Health and Associations with Coping Styles: Replication and Extension of Previous Studies"

_behavsci, 2025, doi:10.3390/bs15040487_

Round 1

Reviewer 1 Report

Comments and Suggestions for Authors

The paper presents a well-structured and methodologically sound study on the psychological impact of content moderation. The research is relevant, well-grounded in existing literature, and contributes valuable empirical insights to the field. However, some areas require further refinement. The following section outlines the key strengths and areas for improvement.

Strengths

The study addresses a significant yet under-researched issue, thus providing a valuable contribution to both theoretical and practical discussions on workplace mental health. The methodological rigour of the study is evident in the use of validated psychological scales, a high response rate, and multivariate analysis, all of which enhance the reliability of the findings. The paper is well-structured, with a logical progression from descriptive to inferential statistics, ensuring clarity and ease of interpretation. The discussion maintains a balanced perspective, avoiding an exaggerated presentation of the results and demonstrating a conscientious acknowledgement of the limitations of self-reported data and the constraints of a cross-sectional design. The paper is characterized by a formal, neutral and professional academic tone, which serves to reinforce its credibility and enhance its readability.

Areas for Improvement

  1. Explicit Formulation of Hypotheses

The hypotheses are implied but not explicitly stated in the introduction. Clearly defining them would improve clarity and allow for a more structured discussion of results. State hypotheses explicitly in the Introduction (p. 3) to guide the reader through the expected relationships between variables (e.g., “H1: Higher exposure will be positively correlated with psychological distress”).

  1. Integration of Theoretical Frameworks

The paper lacks an explicit discussion of theoretical models that could contextualize the findings. Frameworks such as the Job Demands-Resources (JD-R) model (e.g., Bakker & Demerouti, 2007) or Lazarus & Folkman’s Transactional Model of Stress (e.g., Lazarus & Folkman, 1987) would strengthen the interpretation of the results. Integrate a brief discussion of these models and refer back to them in the discussion to provide a stronger theoretical foundation.

  1. Explanation of Conflicting Findings

The study's findings reveal an absence of a statistically significant relationship between exposure to distressing content and overall well-being, which stands in contrast to the findings of previous research. However, this discrepancy is merely cursorily addressed in the study, leaving a more thorough examination of potential explanations, such as differences in sample characteristics, cultural influences, and resilience factors, unaddressed. An in-depth discussion of these factors would serve to enhance the robustness of the analysis and provide a more comprehensive understanding of the relationship between exposure to distressing content and overall well-being.

  1. Implications for Practice and Theory

The discussion of workplace well-being services is insightful but could benefit from more concrete recommendations for intervention strategies. Additionally, the theoretical implications are not fully explored. Include specific recommendations for improving well-being programs and discuss how findings might refine existing stress and coping models.

  1. Justification for Cross-Sectional Design

The study employs a cross-sectional rather than a longitudinal design, yet there is no justification for this choice. Given that psychological distress and secondary trauma can develop over time, a longitudinal approach would provide stronger causal insights. Briefly justify the use of a cross-sectional design and acknowledge its limitations more explicitly. Consider suggesting future longitudinal research to track changes over time.

  1. Consideration of Cultural and Organizational Differences

The sample is predominantly composed of content moderators from Asia and North America; however, the potential cultural and organizational influences on coping strategies and well-being outcomes remain unaddressed. A brief discussion is warranted regarding how cultural differences in stigma, coping mechanisms, and mental health support systems may have influenced the results. If pertinent literature is available, comparative studies on workplace mental health across different cultural contexts should be cited.

  1. Definition and Differentiation of Emotional Coping Strategies

The study categorizes emotion-focused coping as maladaptive, but not all emotion-focused strategies are inherently negative (e.g., cognitive reappraisal can be beneficial, whereas rumination is not). Differentiate between adaptive vs. maladaptive emotion-focused coping and integrate literature on the role of emotional regulation strategies in workplace stress management.

  1. Strengthening the Practical Implications Section

The study offers general recommendations for workplace mental health interventions; however, it does not provide specific, evidence-based solutions that companies could implement. To expand the discussion, targeted interventions should be suggested, such as cognitive behavioral training for moderators, resilience-building programs, or AI-assisted content filtering to reduce exposure to harmful material.

The present study makes a valuable contribution to the research field of content moderators' mental health by offering empirical evidence on stress, coping mechanisms, and workplace interventions. Addressing the aforementioned areas will serve to strengthen the paper's theoretical foundation, clarity, and practical impact. The suggested improvements will enhance readability, ensure a more robust discussion of findings, and make the study's contributions clearer. Good luck.

Author Response

Many thanks for your helpful responses. Document shared with responses in the attached. 

Reviewer 2 Report

Comments and Suggestions for Authors

I have some questions and/or suggestions to improve results presentation:

  • Rows 203-206: I think can be useful add the values of Spence et 203 al., (2024a, 2024b) for each demographic characteristic examined, quote of males, mean of age and so on.
  • Rows 212-214 you write “The Core-10 was demonstrated a significant negative relationship with wellbeing (r = -.57, p < .001) and significant positive relationship with secondary trauma (r = -.61, p < .001).” You report the positive correlation with a minus sign by mistake.
  • The reference to Figure 1 in row 222 is a mistake because it is related to the first MGLM analysis presented after. Remove it.
  • Put the reference to Figure 1 after MGLM1 in row 230 instead of Figure 2.
  • Add degrees of freedom for test F
  • Please use the same label for variables in figures/tables and in the manuscript
  • Figure 1: move the label “Frequency of Exposure” from the label of horizontal axis to the label of colour
  • Remove MGLM2 from Table 2 and report F test in the manuscript as you made for MGLM1.
  • I don’t understand if the results reported in Table2 are obtained with a single analysis or with different analysis for each predictor variable reported in rows. I think for coping styles should be done only one analysis, a multiple regression model. Please add standardized regression coefficients to evaluate the effect size of each relationship in the table and the quote of explained variance for psychological distress, secondary trauma and wellbeing.
  • In figures 3-4-5 it is difficult to distinguish the different data series. I think you could remove them.

Author Response

Thank you very much for your comments. Please find attached. 

Reviewer 3 Report

Comments and Suggestions for Authors

For the authors’ convenience, I have roughly divided my review into the sections the authors themselves use in their paper. I would also like to note right away that what I am suggesting is optional unless stated otherwise; everyone has word limits to work within and around, and I have no way of knowing what other reviewers will request, so please take everything here with a grain of salt and keeping the publication limits in mind. Finally, I review as I read, so if I mention something earlier that you mention later, please simply take it as a sign that as a reader, I would have liked to know that information earlier in the article to boost my comprehension.

Introduction: I really enjoyed reading this introduction. I can see most of the ‘big names’ covered in terms of content moderation research, at least on the non-computational end of things (i.e., pure AI content moderation), and everything is explained clearly and succinctly while making a compelling argument for the research. As someone who has a background in Psychology but now works in Communication, I am really pleased to see specialists who are qualified to properly take into account psychological variables begin to cover this topic in greater depth, as many of us are lacking in clinical experience (research or otherwise) to do this kind of work safely. One thing that is missing is an explicit research question and/or hypothesis, but I suspect this is because it is a replication study; if there is space, it would be great to add in Spence's specific results in bold so we know what to expect with your own results. I look forward to reading the rest!

Materials and Methods: A minor edit here I’d like to request – what were participant averages (and standard deviation) on these scales, and what were the reliability scores (Cronbach’s α)? Another small thing, and I’m not sure if you can disclose it, but is it possible to say in which geographic region the members surveyed were located? I only ask as I myself am located in a region where English is not the dominant language, and so many of my surveys have to be translated, and when they are not, I usually have to confirm English proficiency. If there’s any way to confirm that the respondents either work in an English-language region/company somehow without doxing your collaborator, that would be much appreciated (I just read the next sentence and it looks like they were in Asia, so maybe there’s another way you can confirm English proficiency?).

Results: I see that you’ve got lots of percentages here, but it would still be nice to have some averages too, space allowing. Were the terms “hate speech” and CSAM defined for the population, or did you use internal company guidelines for these? Also, perhaps I missed them, but are there any statistics to show associated with those Bonferroni post-hoc tests? I wasn’t able to find them myself. Although it’s outside of the scope of the present work, if it isn’t already suggested, I would love to see some future qualitative work asking for more depth on the result that customer safety positions seem to have worse perceived mental health outcomes than content moderators! That’s fascinating, and very against any literature I’ve come across or produced myself.

Discussion: Non-clinical core-10 studies may be lacking, but I am very pleased to see one here! I suspect that this population may be more clinical than we often give them credit for, despite companies’ best (or non-existent, depending on the company) efforts. If there is space, I would recommend adding a brief section in this discussion about Asian cultures and their experiences with mental health services, as I believe this could be a major contributing factor regarding your differences in results with Spence. As someone currently based in Asia myself, they are at a very different place in their journey policy-wise and societally in terms of acceptance and understanding than many countries in the West (including the UK). Corporate structures here are also extremely hierarchical on the whole, so if they are asked if something their company is doing is helpful, there is a much higher likelihood that they will rate it as positive, even anonymously. Again, space-allowing, this could inspire some more work over here in this field, which I would argue is sorely needed for the reasons I just outlined.

Author Response

(The authors gave the same response as above.)
